# Modified Polyethylene Foam for Critical Environments

**DOI:** 10.3390/polym14214688

**Published:** 2022-11-03

**Authors:** K. A. Ter-Zakaryan, A. D. Zhukov, I. V. Bessonov, E. Y. Bobrova, T. A. Pshunov, K. T. Dotkulov

**Affiliations:** 1TEPOFOL Ltd., Shcherbakovskaya Str. 3, 105318 Moscow, Russia; 2Department of Building Material Science, National Research Moscow State University of Civil Engineering (NRU MGSU), Yaroslfdskoye Str. 26, 129337 Moscow, Russia; 3Research Institute of Construction Physics of the Russian Academy of Architecture and Construction Sciences, Locomotive Passage, 21, 127238 Moscow, Russia; 4GASIS Institute, National Research University Higher School of Economics (NRU HSE), 20 Myasnitskaya Str., 101000 Moscow, Russia

**Keywords:** modified polyethylene, polyethylene waste, foamed polymers, combustibility, operational stability, visualisation

## Abstract

One of the most important priorities for all countries with property beyond the Arctic Circle and territories located in permafrost areas is the development of special construction technologies and systems. The required conditions are met by insulation systems based on seamless insulation shells made of polyethylene foam. The study of the strength and performance properties of polyethylene foam and its combinability was carried out according to standard methods and using the methods of experimental design and the analytical processing of the results. The change in material properties at negative temperatures was determined based on the results of climatic tests, followed by an evaluation of creep under load. The evaluation of the effectiveness of the design solutions was carried out using special computer programs. It was found that the performance characteristics of products made of polyethylene foam (rolls, mats) meet the requirements for insulation materials used at temperatures down to −60 °C. The resulting material is moderately combustible, which must be taken into account when developing recommendations for its use in insulation systems. A nomogram has been developed that makes it possible to predict the properties of a material and solve formulation problems. Insulation systems were developed, and a visualisation of the thermal fields of the insulation systems of the external walls and ceilings of a building was carried out.

## 1. Introduction

Foamed polymers have low thermal conductivity and vapor and water permeability; the stability of the properties over time and in varying humidity makes them promising for use as thermal insulation in structures in contact with the ground and in environments with high humidity. Expanded polystyrene and polyurethane-foam-based products are widely used in construction, and the applications of polyisocyanurate foam and foamed polyolefins, including polyethylene foam, are expanding [1,2,3].

The range of applications of foamed polymers can be expanded if formulations and technologies are developed that reduce the flammability of such materials and increase their resistance to negative and changing temperatures. It is also important to introduce technologies that use secondary products based on processed polymer waste [4,5,6].

An additional impetus for the expansion of the use of foamed plastics comes from the orientation of developed and developing economies towards the development of northern territories: Russia, Canada, the USA, Denmark (Greenland). This becomes particularly relevant against the background of global warming, which makes the climate in the polar regions more conducive to life. If we consider Russia, the prospects for the development of the northern regions are obvious. They are, first, the development of the Northern Sea Route; second, the expansion of opportunities for the extraction of liquid and gaseous hydrocarbons; and third, the expansion of the extraction of minerals and, in particular, bauxite [7,8,9].

Building in the Arctic poses additional risks. The most important is the presence of permafrost, the preservation of which is a prerequisite for maintaining the bearing capacity of the foundation for any structure. It should be noted that the frozen condition of the ground can be disturbed by both human and natural causes. Natural causes include increases in average annual air temperatures and precipitation. Technogenic causes include the exposure of the soil to heat emanating from operating equipment. Protection against the penetration of dripping wetness from rain or meltwater into the soil is possible through special design solutions. The design of the insulating envelope of structures includes the reduction of heat losses to the environment and is aimed both at increasing the comfortable conditions on the premises and reducing the negative impact from the frozen ground [10,11,12].

As mentioned above, the properties of foamed polymers fully meet the requirements of polar regions. At the same time, it is necessary to solve the problems related to the reduction of fire hazard and the possibility of waste disposal, as well as the structural problems caused by the placement of foamed polymers in structures and the solution of the problem of connections [13,14,15].

It should be noted that the joints between individual thermally insulating panels, as well as the panels and elements of the supporting structures, are heat-transfer bridges (both conductive and convective, due to air filtration through leaks), and this process is exacerbated with an increase in the temperature difference between the outside and inside temperatures. The most promising in this respect is the use of rolled products based on foamed polyethylene foam. The technology developed by TEPOFOL LLC makes it possible to obtain seamless insulation shells with practically no “cold bridges” (RF Patent No. 2645190). Polyethylene rolls are mechanically attached to the structure and joined to form a lock. The lock joint is welded with hot air, creating a seamless insulating shell [16,17].

Foamed polyethylene is a gas-filled thermoplastic polymer, a soft and elastic material with shock-absorbing properties. Various additives can be added during the technological processing of the material. The chemical and thermal resistance of polyethylene foam is improved by chlorination, bromination and fluorination. To reduce aging under the influence of sunlight and ultraviolet radiation, carbon black and benzophenone derivatives are added.

It is promising to use recycled polyethylene for the production of polyethylene foam. The change of properties is carried out by introducing cross-linking and foaming reagents (dicumyl peroxide and azodicarbonamide, their analogues with lower toxicity, as well as azodicarbonamide, etc.). The introduction of mineral and (or) organic fillers is practised [18,19,20].

Reducing the flammability of foamed polymers is done by chemical or physicochemical methods. Freons, propane–butane or isobutane are used to foam the polymer matrix, and the presence of oxygen is possible. By replacing the foaming gases with non-flammable gases, the flammability of the material can be reduced. The introduction of finely dispersed (or ultradisperse) mineral powders containing intracrystalline or interpacket water (e.g., magnesium hydroxide or some types of clays) into the matrix also helps reduce combustibility but may have a negative effect on the properties of the material.

The chemical manipulation of the polymer matrix by substances that reduce the flammability of the polymer occurs at the stage of the polymerisation of the starting product, including the use of secondary raw materials. In this context, the studies conducted at NRU MGSU on the modification of cellulose fibers are of interest. Studies investigating the possibilities of modifying the properties of cellulose fibers used as a basis for a group of heat-insulating materials and as a reinforcing component for construction products have shown the feasibility of using borates and their derivatives [21,22].

The modification of the plant material of the product with a composition based on monoethanolamine (N→B) trihydroxyborate consists of the depolymerisation of lignin and hemicellulose. The modifier molecules are located in the lignocarbohydrate complex of the cell walls. This leads to an increase in the content of the boron–nitrogen compound in the composition of the cell walls and, consequently, to a decrease in the combustibility of the polymer. The modification process is enhanced by increasing the temperature and under drying conditions [23,24].

Rolled polyethylene foam and seamless trays based on it have already proven their worth in cold-storage applications. In modern ice palaces (Figure 1), the insulating shell helps create a comfortable mode for users and reduces energy costs for maintaining the room temperature and ice in an operational state. A reduction of heat loss through building elements due to the use of a seamless insulation shell with mechanical fastening significantly reduces the load on the artificial air-conditioning equipment.

The task of maintaining cold is also implemented in the technology of snow preservation in the “off-season” (from spring to autumn). In this case, a seamless insulating shell based on polyethylene foam performs the function of protection from the heat of the environment. The presence of preserved snow in early autumn allows opening the ski season in mountain areas one month earlier, which increases the profitability of mountain-sports tourism [25,26].

Rolls of polyethylene foam were prepared and laid out on an insulated surface and then joined with an interlocking joint, the joints being monolithic. A wrapped polyethylene foam was used, 50–100 mm thick and with a reflective coating. To increase the operational stability of the insulation system and ensure the possibility of reuse, an awning cover is applied. The coating is fixed by means of cables placed along the perimeter of the insulated area (Figure 2).

The metallised surface of the rolled polyethylene foam and the light-coloured awning ceiling reflect the sun’s rays. The low thermal conductivity of the insulation shell, the stability of its properties and the minimal heat loss at the joints make it possible to achieve a thermal resistance of the insulation shell (“thermal blanket”) in the order of 1.3–1.6 m^2^ °C/W. The technology for the protection and preservation (conservation) of snow cover was first implemented in 2013–2014 on the mountain slopes of the Rosa Khutor Olympic area in Sochi. By the start of the 2014 Winter Olympics, about half a billion cubic metres of snow mass had accumulated here. To date, this technology has been used several times, confirming the feasibility of using polyethylene-foam-based shells under negative and changing temperature conditions.

The aim of the research presented in this article was to develop a method for selecting a composition of modified polyethylene foam, to conduct studies to assess the flammability of the materials obtained, to evaluate the properties of polyethylene foam at low temperatures, and to develop insulation systems for building structures for operation in conditions typical of the Arctic and Antarctic regions and in the presence of permafrost soils.

## 2. Materials and Methods

Foam polyethylene was traditionally used in structures as a lining or vibration-damping material in rolls or sheets whose thickness did not exceed 5 mm. TEPOFOL technology makes it possible to obtain products with a thickness of 5 to 200 mm, which greatly expands the possibilities of using polyethylene foam. 

The preparation (processing) of secondary polyethylene includes the following technological steps: collection and storage, grinding and agglomeration, removal of foreign inclusions, introduction of modifiers, including flame retardants, granulation. If waste from own production (offcuts, low-quality products) is used, it is granulated after being processed by pyrolysis and the addition of modifiers and then dosed into the feed hopper of the extruder. In the extruder, the polyethylene granulate is melted and the melted mass is mixed under pressure with the foaming agent isobutane. The continuous mixing gives the polymer mass a homogeneous structure, also at the molecular level. The cell size of polyethylene foam is up to 1.2 mm. In the lamination method, the finished polyethylene foam rolls are coated on one or both sides with a heat-reflecting layer of foil or metallised lavsan. To achieve the desired thickness, the layers of foamed polyethylene are welded together by “heat welding”.

Originally, the products were monolithic, with welded layers (each 10 mm thick) with or without a heat-reflective coating (Figure 3a). Since polyethylene foam not only has low thermal conductivity but also low vapour and water permeability, as well as high moisture resistance and operational stability in contact with the ground, it became possible to produce seamless insulation shells for various building structures with high insulating capacity.

The next step in the development of technologies was the creation of a new range of air-layer materials (RF utility model protection no. 199048) [27,28]. These are insulating multilayer materials containing flat layers of foamed plastic (polyethylene, polypropylene or rubber) joined together by seams, with air gaps formed between the layers (Figure 3b). Such a system has lower thermal conductivity than monolithic layered materials, and insulation shells have better thermal performance.

The evaluation of the strength properties of polyethylene foam, patterns of moisture exposure, operational stability and thermal conductivity was carried out according to the methods specified in state standards and norms. The research was carried out at the Moscow Building University (NRU MGSU), the Institute of Building Physics (NIISF RAASN), in the scientific departments of TEPOFOL LLC. Part of the research presented in this paper was carried out using the equipment of the Central Centre for Collective Use, named after Prof. Yu.M. Borisov VSTU, supported by the Ministry of Science and Higher Education of the Russian Federation, Agreement No. 075-15-2021-662.

The composition of recycled polyethylene and the intervals of change in the consumption of the flame retardant and modifier were determined on the basis of the analysis of a priori information and a preliminary series of experiments. Based on the analysis of scientific sources (articles in scientific journals), the factors influencing the reaction functions and the intervals of their change in the experiment were determined.

A preliminary set of experiments was carried out (using the fractional replicate method), which made it possible to identify the most important factors and clarify the intervals int he variation of these factors.

As a result of the analysis of the literature and preliminary studies, the reaction functions and the factors varying in the experiment were determined: the consumption of recycled polyethylene (X_1_), the consumption of the modifier (X_2_) and the consumption of the flame retardant (X_3_). Maleic anhydride and boron nitrogen compounds were used as modifiers. The flame retardant (magnesium oxide hydrate) and the modifier were added to the recycled polyethylene blend during heat treatment and subsequent granulation. 

The experiment to determine the optimal composition of modified polyethylene foam was conducted using mathematical design methods based on complete D-optimal schedules with three factors, followed by the testing of statistical hypotheses and the analytical optimisation of the results.

The response functions are: the average density of the polyethylene foam products (Y_1_, kg/m^3^) and the compressive strength of the samples (cube 100 × 100 × 100 mm^3^) at 10% deformation (U_2_, kPa). The compressive strength of the samples was used as an optimisation parameter. The accepted reaction functions: compressive strength at 10% deformation (Y_2_) and average density (Y_1_) are the main normative properties of both polyethylene foam and other foamed plastics (polystyrene foam, polyurethane foam, polyisocyanurate foam). The strength properties determine the operational stability of the products, and the average density of foamed plastics is a property that correlates with the thermal conductivity coefficient. The conditions of the experiment and the variation intervals of the factors are listed in Table 1.

Analytical optimisation [29]. It consisted of determining the extrema of the response functions (Y_1_ and Y_2_). The response functions were considered mathematical polynomials, each containing three variables. The optima of these functions were determined in the ranges of change of each factor (variable) and at the limits of the ranges. It was taken into account that the accuracy of the prediction of the result in the central region is higher than in the peripheral regions. The solution of polynomials taking into account the extreme value functions found made it possible to obtain optimised regression equations for one or more variables.

Tests on the flammability of polyethylene foam with an optimised composition were carried out according to the standards of the “Method for testing combustible building materials to determine their flammability groups”. The flue gas temperature, the duration of spontaneous ignition and (or) smoldering, the damage length and the change in mass, as well as the appearance of the samples after the test, were determined.

The evaluation of the developed insulation systems was carried out with the help of a graphical visualisation. The visualisation of design solutions for insulation systems was carried out at the Research Institute for Building Physics using special computer programs.

The studies were conducted in accordance with state standards on the territory of the Russian Federation and CIS, most of which are updated with international standards.

## 3. Experimental and Results

The regression equations (base polynomials) were obtained as a result of the statistical processing of the experimental results. The significance of the coefficients of the regression equations was assessed using the confidence interval determined in accordance with the values of the Student’s test (*t*-test) and the variance of the parallel experiments. The values of the confidence intervals are the same: for the polynomial of the average density Δb_1_ = 1.2 kg/m^3^ and for the polynomial of the strength Δb_2_ = 9 kPa.

After checking the significance of the coefficients (coefficients identified as insignificant by the absolute value of smaller confidence intervals and equated to zero), dependencies were determined that established a functional relationship between the variable factors and the resulting parameters:for average density (with a confidence interval Δb_1_ = 1.2 kg/m^3^):
Y_1_ (kg/m^3^) = 25.3 + 4.1X_1_ − 4.8X_2_ + 2.1X_3_ − 1.9X_1_X_2_(1)

for compressive strength (at a confidence interval Δb_2_ = 9 kPa):

Y_2_ (kPa) = 134 + 33X_1_ + 24X_2_ + 18X_3_ + 14X_1_X_2_ − 11X_3_^2^(2)

Further, according to Fisher’s criterion, a check was carried out for each model (polynomial), which showed the adequacy of the models obtained. The evaluation of the adequacy of the obtained models (regression equations with only significant coefficients) was done by comparing the calculated values of the Fisher criterion (F criterion) with its table values, which are 19.2. The calculated values of the F-criterion as the ratio of the variance of adequacy to the variance of the parallel experiments are the same: for the polynomial of average density 14.2 and for the polynomial of strength 13.7.

Analysing the magnituds and signs of the coefficients of the regression equations, one can determine the degree and direction of the influence of each factor on the result. The graphical interpretation of the obtained dependencies is shown in Figure 4 and Figure 5.

The average density of polyethylene foam (Y_1_, kg/m^3^) depends most on the consumption of the modifying additive; moreover, the density of polyethylene foam decreases with the increasing consumption of the additive (the coefficient at “X_2_” is equal to “−4.8”). This is understandable from the point of view of the mechanism of action of maleic anhydride, which, under the conditions of the melting of the polymer in the extruder, exerts a plasticising effect on the melt, thereby reducing the surface tension, which leads to the better swelling of the mixture.

An increase in the consumption of secondary polyethylene additives (the coefficient at “X_1_” is equal to “+4.1”) leads to a slight increase in the average density of the polyethylene foam, which is due to an increase in the melt viscosity in the extruder. An increase in flame-retardant consumption leads to a slight increase in the average density coefficient at “X_3_” equal to “+2.1”, which is due to some reduction in the effect of melt foaming in the presence of a finely dispersed mineral component.

The compressive strength of the sheets at 10% deformation (Y_2_, kPa) depends most on the consumption of recycled polyethylene (the coefficient at “X_1_” is equal to “+33”); the flow rate of the modifying additive affects the result to a lesser extent. Moreover, an increase in the values of each factor leads to a greater or lesser increase in strength (the coefficient at “X_2_” is equal to “+24”). The direct influence of the flame retardant on the strength is the least significant of all factors (the coefficient at “X_3_” is equal to “+18”). The positive values of the coefficients indicate that an increase in the values of each of the variable factors (or cost in physical terms) in the intervals on which the experiment is based leads to an increase in the strength properties of polyethylene foam.

The combined effect of the consumption of recycled polyethylene and the modifier on the change in average density is small (the coefficient at “X_1_X_2_” is equal to “−1.9 kg/m^3^” in the polynomial for average density and “14 kPa” in the polynomial for compressive strength at 10% elongation). Nevertheless, the recorded fact of the joint effect of the consumption of recycled polyethylene and the modifier on density and strength should be additionally studied to identify possible synergistic or antagonistic effects.

A peculiarity of the polynomial Y_2_(X_1_, X_2_, X_3_) is that the dependence of the compressive strength of the plates at 10% deformation (Y_2_, kPa) on the consumption of flame retardant (X_3_) is not linear (the coefficient at “X_3_^2^” is equal to “−11”). This means that, when the consumption of a flame retardant increases (ceteris paribus), the strength first increases and then begins to decrease. Using the method of analytical optimisation, you can determine the range of the consumption of modifiers (X_3_) in which the strength is maximum.

## 4. Discussion

The function Y_2_(X_1_, X_2_, X_3_) has a local optimum with respect to the factor X_3_, determined by the differential analytic method. The effect of flame-retardant consumption on strength is extreme, so it is possible to optimise the algebraic functions of the three variables (1) and (2) by the factor X_3_ in the course of the investigation.

The optimisation by the factor X_3_ (flame-retardant consumption) is performed in the following order: by differentiation, and the optimal flame-retardant consumption is determined in coded values (reduced to the interval [−1; 1]) and in natural values. Then, Equations (1) and (2) are solved with the determined optimal value of the factor X_3_ (in coded form), and functions (3) and (4), optimised for X_3_, are obtained. 

We determine the optimal consumption of the flame retardant in its natural form (using the data in Table 1): Ca = 9 + 0.82 × 4 = 12.3%(3)

We calculate the X_3_-optimised response functions: for average density (with a confidence interval Δb_1_ = 1.2 kg/m^3^):
Y_1_ = 25.3 + 4.1X_1_ − 4.8X_2_ + (2.1 × 0.82) − 1.9X_1_X_2_(4)

for compressive strength (at a confidence interval Δb_2_ = 9 kPa):

Y_2_ = 134 + 33X_1_ + 24X_2_ + (18 × 0.82) + 14X_1_X_2_ − (11 × (0.82 ×2))(5)

We get the X_3_-optimised response functions:for average density (with a confidence interval Δb = 1):
Y_1_ = 27.0 + 4.1X_1_ − 4.8X_2_ − 1.9X_1_X_2_(6)

for compressive strength (at a confidence interval Δb = 8):

Y_2_ = 142 + 33X_1_ + 24X_2_ + 14X_1_X_2_(7)

Graphical interpretation of dependences (6) and (7) is shown in Figure 6 and Figure 7.

Considering the optimisation data and using the optimised functions Y_1_(X_1_, X_2_) and Y_2_(X_1_, X_2_), a nomogram was constructed (Figure 8). Using this nomogram, it is possible to solve the problem of predicting properties (compressive strength at 10% deformation and average density) depending on the consumption of recycled polyethylene and the pressure in the extruder, as well as the direct problem of choosing the consumption of recycled polyethylene under the condition of the given properties.

The evaluation of the properties of the modified polyethylene foam was carried out on samples of 100 × 100 × 100 and 500 × 500 × 50 mm^3^, the composition of which was selected taking into account the analytically determined optimum flame-retardant consumption (11.7 kg/m^3^) and plasticiser content (6.8% by weight of the polymer). The recycled polyethylene content was set at 23% of the polymer weight according to the nomogram (Figure 6), and the gas pressure in the second extruder was kept at a level of 90–91 kPa. CO_2_ was used as the foaming gas.

It was found that the compressive strength of products at 10% deformation is 140–160 kPa at an average density of 22–24 kg/m^3^ and depends on the loading area and can be 260 kPa for loading areas larger than 100 m^2^. The tensile strength in the longitudinal direction is 80–92 kPa, and the weld strength is 29–32 kPa. Polyethylene foam has a thermal conductivity of 0.032–0.034 W/(m∙K); the diffusion moisture absorption without metallised coating is 0.44 kg/m^2^ and with a coating is 0.37 kg/m^2^; water absorption, when partially immersed in water for 24 h, is 0.013 kg/m^2^; water absorption by volume when fully immersed in water for 28 days is 0.96%. The material practically does not change its properties under the conditions of long-term, alternating temperature fluctuations from −60 to 70 °C.

When evaluating the flammability of polyethylene foam samples with an optimised composition, the following results were obtained: the flue gas temperature was 220–230 °C; the duration of self-smoldering was 15–20 min; the damage length of the sample was 10–14%, the weight loss was 10–15%; the formation of a burning melt was not detected. Studies to assess the degree of flammability and other standardised properties: flame spread over the surface, smoke development and the toxicity of combustion products continue in certified laboratories.

The properties of modified polyethylene foam made from recycled polyethylene are not fundamentally different from the properties of polyethylene foam, which is produced from “pure” raw materials. Taking into account the compressibility and strength properties of the material, one of the possible applications is its use in seamless insulation shells for an apartment roof under a concrete screed or its use in insulation systems for pilaf in contact with the ground.

From a construction point of view, the development of the northern areas involves the implementation of the following tasks: the construction of buildings and structures that provide heat preservation, comfortable indoor conditions and the possibility of carrying out technological processes (for which positive temperatures are also important); protection of foundations; and taking into account the preservation of permafrost.

Materials for the insulation systems of buildings and structures for northern latitudes must have not only low thermal conductivity but also operational stability, as well as the stability of properties under difficult conditions of use. Also important is the vapor and water permeability of such materials, as well as resistance to aggressive environments, including the effects of groundwater. These requirements are met by the roll products made of foam polyethylene.

The thawing of frozen ground occurs through the radiative effect of solar heat, as well as through heat loss from poorly insulated buildings. A major threat to frozen ground is water (rain, melt). The destruction of permafrost when in contact with such water occurs much more rapidly than when exposed to radiation. Systems for ground insulation along the perimeter of buildings on piles are currently being developed (Figure 9).

In this system, the soil is removed along the perimeter of the building in a strip 1.5 m wide to a depth of 0.5 m. The soil is then filled with earth. The pit is then backfilled with soil. In this system, the polyethylene foam has, firstly, the function of cutting off the melting and rain water that forms near the main building, and, secondly, it is a heat-shielding barrier that prevents the penetration of external heat into the frozen ground.

As practise shows, in winter conditions, when temperatures can drop to −50 °C or less and significant wind loads occur, there is considerable heat loss, even in insulated buildings. The decisive factor is the formation of cold bridges at the contact points between the thermally insulating panels and between the structural elements of the building. A rolled material with a reflective coating not only adds thermal resistance to the structure but also completely isolates the areas of convective heat transfer at the joints (filtration and suction of cold air from the environment). Schemes of overlapping a residential building without insulation with rolled polyethylene foam and with insulation can be found in Figure 10.

Rolled polyethylene foam with a reflective coating (foil or metallised) also forms a protective layer along the outer perimeter of the structure, and inside is the basis of a floating floor, which is laid on the slabs of the supporting structure.

The visualisation was performed with the computer program THERM. This program allows you to create models of two-dimensional heat transfer in building elements. Analysis of heat transfer with the program allows you to evaluate the energy efficiency of the structure and local temperatures of the sample and solve problems with condensation and the moisture of the material of the structure and its tightness.

The computer program THERM was used to simulate the conditions for modelling two-dimensional heat transfer in the enclosing structures of a pile building (Figure 11). The construction of such buildings is practised on frozen soils.

The modelling shows that, at the points where the structure rests on the column, due to the high thermal and temperature diffusion capacity of the supporting structures, a “temperature bridge” is maintained, which passes through the points of contact of the structural elements (“column-pipe-bearing wall”). At the same time, by placing rolled polyethylene foam on the insulated outer surface (above the cold space under the structure) and as an element of the floating floor inside the room, you can completely block the possible paths of heat loss both through the “temperature bridge” and by blocking the direct penetration of cold air at the joints of the structures.

## 5. Conclusions

Insulation materials based on polyethylene foam are environmentally friendly, easy to use and install and have high operational stability. The use of production waste and recycled polyethylene slightly changes the properties of the material. The developed nomogram allows determining the optimal consumption of recycled polyethylene, taking into account technological factors, and can be used in the development of a methodology for selecting the composition of modified polyethylene foam using recycled polyethylene.

Under critical operating conditions, the shortcomings of structures and materials that are hardly noticeable when used in mild climates become more noticeable. One is the potential for cold-air infiltration through leaks in the structure and insulating envelope, and the other is the presence of cold bridges. When constructing and operating buildings and structures on permafrost, keeping the ground in a frozen state is no less important. The development of materials and system solutions has been implemented in these areas.

It was found that the performance of polyethylene foam with the partial use of recycled waste-based polyethylene, as well as insulating shells based on this material, fully meets the requirements for materials and systems used in harsh climatic conditions and in contact with frozen or wet soil. Maintaining the temperature regime in the insulated volumes and during the insulation of the floor is ensured by the effective performance of the material and the possibility of forming a seamless connection between the individual insulating elements.

## Figures and Tables

**Figure 1 polymers-14-04688-f001:**
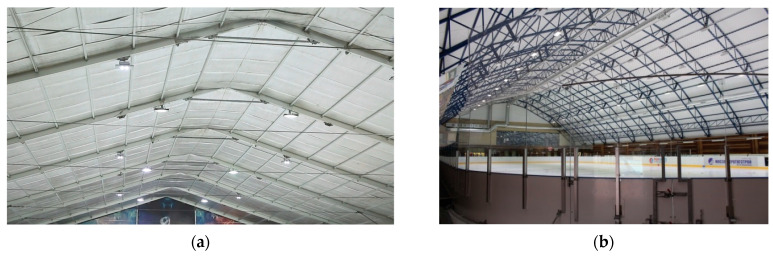
insulation of sports facilities: (**a**) ice rink; (**b**) ice stadium.

**Figure 2 polymers-14-04688-f002:**
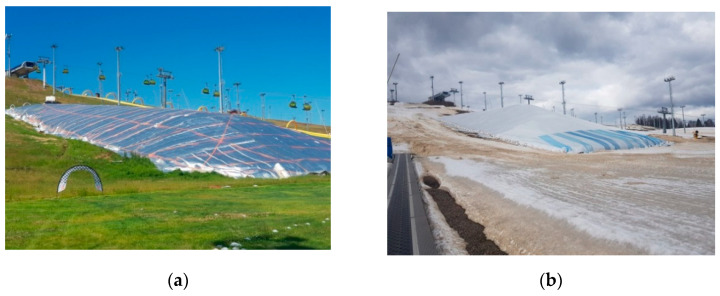
Stored snow reserves under (**a**,**b**) the “thermal blanket”.

**Figure 3 polymers-14-04688-f003:**
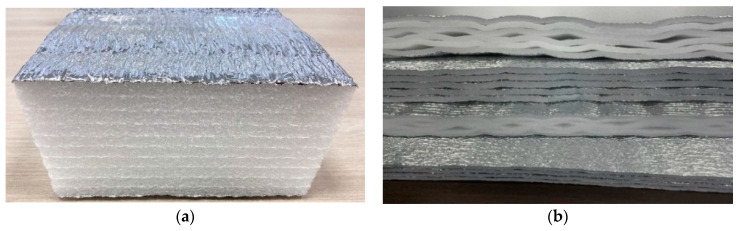
Polyethylene foam products: (**a**) layered products; (**b**) multi-layered products with an air gap from the air layer line.

**Figure 4 polymers-14-04688-f004:**
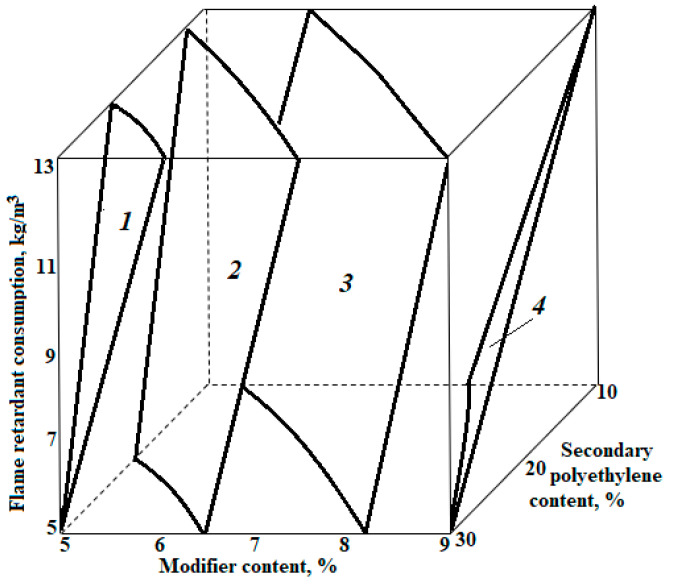
Dependence of the average density of polyethylene foam on variable factors: 1–32 kg/m^3^; 2–28 kg/m^3^; 3–24 kg/m^3^; 4–20 kg/m^3^.

**Figure 5 polymers-14-04688-f005:**
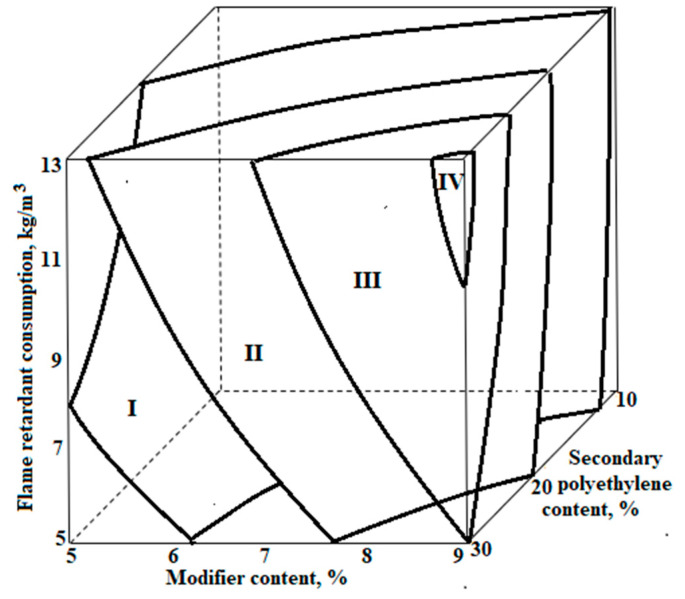
Dependence of compressive strength at 10% deformation of polyethylene foam samples on variable factors: I—200 kPa; II—180 kPa; III—160 kPa; IV—140 kPa.

**Figure 6 polymers-14-04688-f006:**
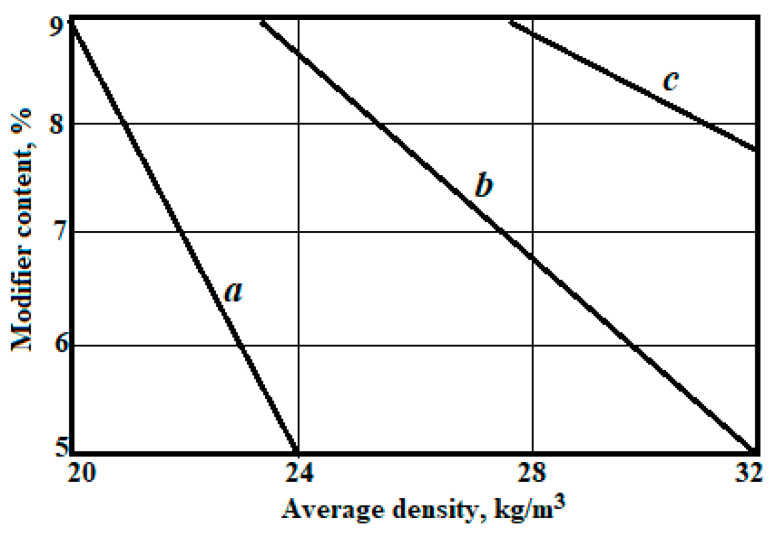
Dependence of the average density of polyethylene foam with optimised-flame retardant consumption on variable factors, with the content of recycled polyethylene: a—10 kg/m^3^; b—20 kg/m^3^; c—30 kg/m^3^.

**Figure 7 polymers-14-04688-f007:**
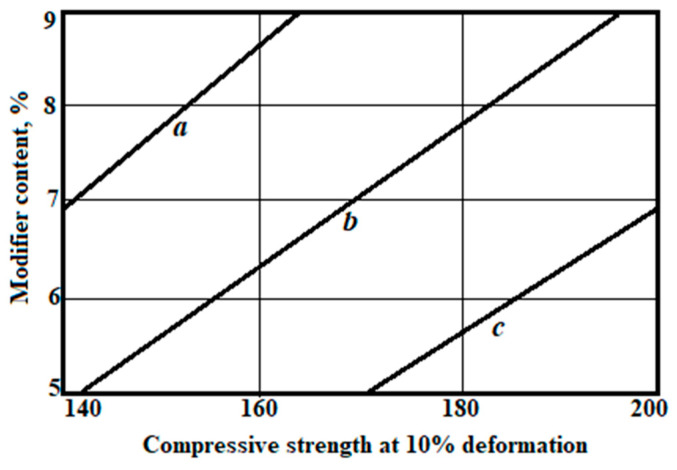
Dependence of compressive strength at the 10% deformation of polyethylene foam samples, with the content of recycled polyethylene content: a—10 kg/m^3^; b—20 kg/m^3^; c—30 kg/m^3^.

**Figure 8 polymers-14-04688-f008:**
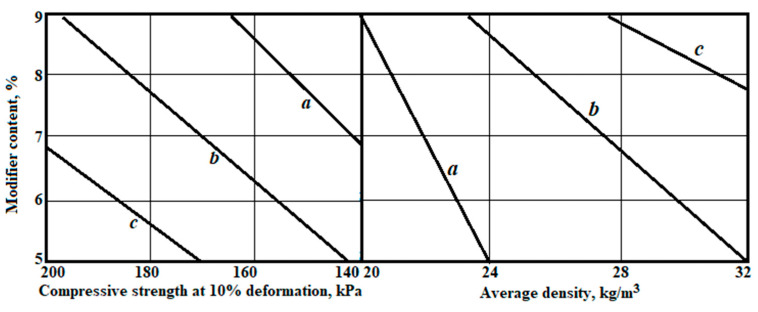
Nomogram for determining the consumption of recycled polyethylene and modifier, with the content of recycled polyethylene: a—10 kg/m^3^; b—20 kg/m^3^; c—30 kg/m^3^.

**Figure 9 polymers-14-04688-f009:**
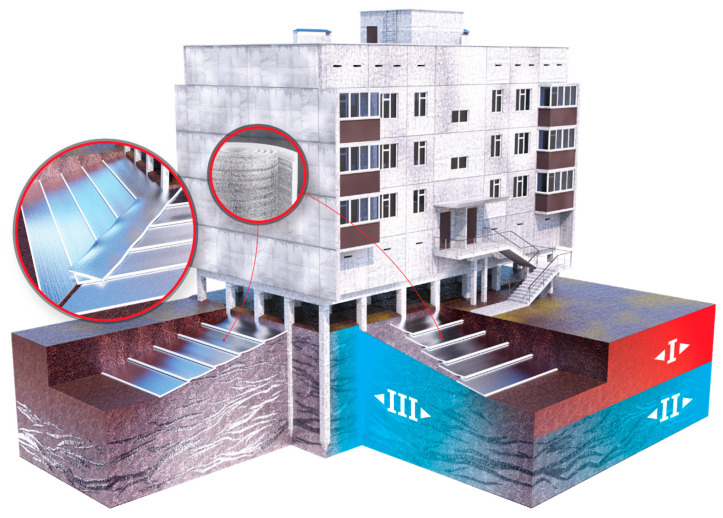
Scheme of insulation along the perimeter of the pile building: I—summer thaw layer; II—permafrost; III—permafrost is preserved in the whole depth.

**Figure 10 polymers-14-04688-f010:**
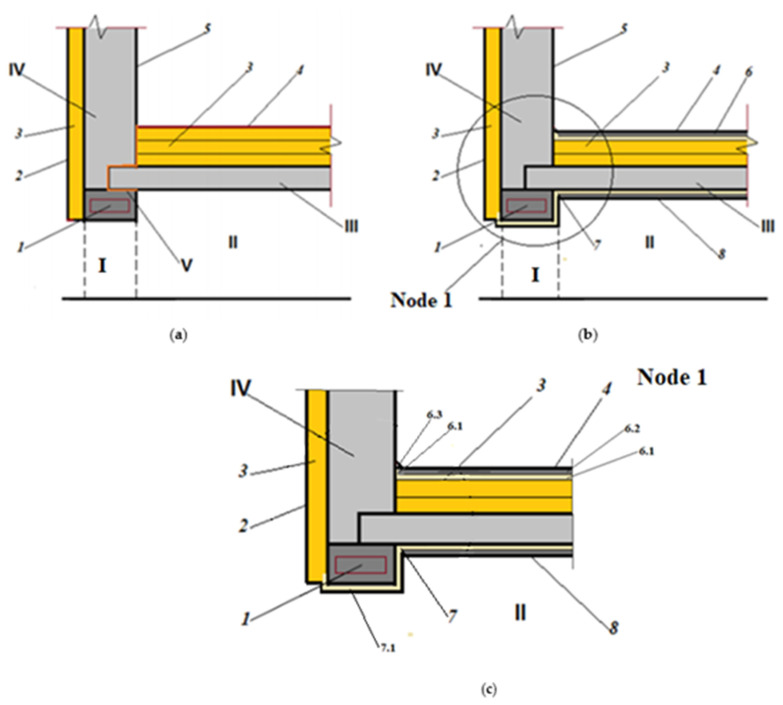
Floor plan of a residential building (Section 2—wall between columns and details of node 1): I—column; II—ventilated space; III—overlap above ventilated space; IV—load-bearing wall; V is the area of increased heat transfer and cold-air infiltration; 1—connection of columns; 2—facade insulation system; 3—thermal insulation of ceiling; 4—floor covering; 5—interior wall cladding; 6—floating floor system (dry assembly), includes: 6.1—PES-20 mm-thick layer laid on the wall; 6.2—10 mm-thick chrysotile cement board, then the floor covering; 6.3—plinth; 7—insulation above a ventilated space; 7.1—insulation up to SFTK (completely covers infiltration (flow through the joint)); 8—protective cladding; (**a**)—insulation with thermal insulation boards; (**b**)—seamless insulation with rolled polyethylene foam; (**c**)—node 1.

**Figure 11 polymers-14-04688-f011:**
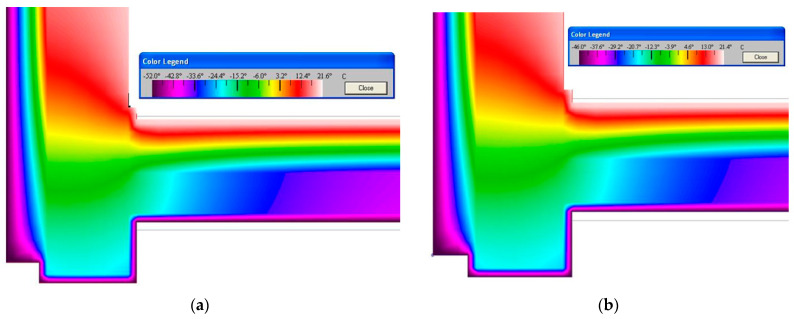
The structure of the formation of the temperature field in a structure isolated above the ventilated space (between the supporting columns): (**a**)—visualisation of the temperature field for Yakutsk (−52 °C); (**b**)—visualisation of the temperature field for Norilsk (−46 °C).

**Table 1 polymers-14-04688-t001:** Experimental conditions.

Factor Name	SymbolX_i_	$\mathrm{The}\;\mathrm{Average}\;\mathrm{Value}\;\mathrm{of}\;\mathrm{the}\;\mathrm{Factor},\;\bar{X_{i}}$	Variation Interval, ΔX_i_	Factor Values at Levels	
				−1	+1
Content of recycled polyethylene, %	X_1_	20	10	10	30
Modifier content, %	X_2_	6	1	5	7
Flame retardant consumption, kg/m^3^	X_3_	9	3	6	19

## Data Availability

Data confirming the reported results can be found on the website of Tepofol LLC.

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
