# Peer review of "Modified Polyethylene Foam for Critical Environments"

_polymers, 2022, doi:10.3390/polym14214688_

Round 1
Reviewer 1 Report
This manuscript reports modified polyethylene foam for critical environments. This research selected the composition of modified polyethylene foam to develop insulation systems for building structures for operation in the presence of permafrost soils. The topic is interesting. However, many points of the manuscript should be improved as follows.
1. The abstract should be concise.
2. The key data about Y1 and Y2 must be included in the manuscript.
3. The method of preparation of polyethylene foam should be added.
4. The reason of choosing content of recycled polyethylene,modifier content and flame retardant consumption should be further discussed.
Author Response
Reviewer 1
1 Abstract should be concise.
"Abstract" edited.
- The key data about Y1 and Y2 must be included in the manuscript.
The accepted reaction functions: Compressive strength at 10% deformation (Y2) and average density (Y1) are the main normative properties of both polyethylene foam and other foamed plastics (polystyrene foam, polyurethane foam, polyisocyanurate foam). The strength properties determine the lifetime of the products and the average density of foamed plastics is a property that correlates with the thermal conductivity coefficient.
- The method of preparation of polyethylene foam should be added.
Technology for the production of non-crosslinked polyethylene foam (NPE)
Polyethylene granules are melted in the extruder and the melted mass is mixed under pressure with the foaming agent isobutane. The continuous mixing gives the polymer mass a homogeneous structure, even at the molecular level. The cell size of polyethylene foam is up to 1.2 mm. In the lamination method, the finished polyethylene foam rolls are coated on one or both sides with a heat-reflecting layer of foil or metallised lavsan. To achieve the desired thickness, the layers of polyethylene foam are joined together by "heat welding".
- The reason of choosing content of recycled polyethylene,modifier content and
flame retardant consumption should be further discussed.
The compositions of recycled polyethylene, the change intervals of the consumption of flame retardant and modifier are determined on the basis of the analysis of a priori information and a preliminary series of tests.
Analysis of scientific sources (articles in professional journals) was used to determine the factors influencing the response functions and the intervals of their variation in the experiment. A preliminary series of experiments was carried out (using the fractional replicate method), which made it possible to identify the most important factors (the recycled polyethylene content; the modifier content, the flame retardant consumption) and to clarify the ranges of variation of these factors. A plan was drawn up for a full three-factor experiment. The matrix of this experiment (the last series) is presented below. The article presents only the results of the last series of experiments.
Rotatable central composition plan and matrix F for n=3
|
Plan elements |
Matrix F |
|||||||||
|
Ð¥0 |
Matrix F |
Ð¥1Ð¥2 |
Ð¥1Ð¥3 |
Ð¥2Ð¥3 |
||||||
|
Ð¥1 |
Ð¥2 |
Ð¥3 |
||||||||
|
The core of the plan |
1 |
-1 |
-1 |
-1 |
1 |
1 |
1 |
+1 |
+1 |
+1 |
|
1 |
+1 |
-1 |
-1 |
1 |
1 |
1 |
-1 |
-1 |
+1 |
|
|
1 |
-1 |
+1 |
-1 |
1 |
1 |
1 |
-1 |
+1 |
-1 |
|
|
1 |
+1 |
+1 |
-1 |
1 |
1 |
1 |
+1 |
-1 |
-1 |
|
|
1 |
-1 |
-1 |
+1 |
1 |
1 |
1 |
+1 |
-1 |
-1 |
|
|
1 |
+1 |
-1 |
+1 |
1 |
1 |
1 |
-1 |
+1 |
-1 |
|
|
1 |
-1 |
+1 |
+1 |
1 |
1 |
1 |
-1 |
-1 |
+1 |
|
|
1 |
+1 |
+1 |
+1 |
1 |
1 |
1 |
+1 |
+1 |
+1 |
|
|
Star points |
1 |
+1,682 |
0 |
0 |
2,828 |
0 |
0 |
0 |
0 |
0 |
|
1 |
-1,682 |
0 |
0 |
2,828 |
0 |
0 |
0 |
0 |
0 |
|
|
1 |
0 |
+1,682 |
0 |
0 |
2,828 |
0 |
0 |
0 |
0 |
|
|
1 |
0 |
-1,682 |
0 |
0 |
2,828 |
0 |
0 |
0 |
0 |
|
|
1 |
0 |
0 |
+1,682 |
0 |
0 |
2,828 |
0 |
0 |
0 |
|
|
1 |
0 |
0 |
-1,682 |
0 |
0 |
2,828 |
0 |
0 |
0 |
|
|
Plan Center |
1 |
0 |
0 |
0 |
0 |
0 |
0 |
0 |
0 |
0 |
|
1 |
0 |
0 |
0 |
0 |
0 |
0 |
0 |
0 |
0 |
|
|
1 |
0 |
0 |
0 |
0 |
0 |
0 |
0 |
0 |
0 |
|
|
1 |
0 |
0 |
0 |
0 |
0 |
0 |
0 |
0 |
0 |
|
|
1 |
0 |
0 |
0 |
0 |
0 |
0 |
0 |
0 |
0 |
|
|
1 |
0 |
0 |
0 |
0 |
0 |
0 |
0 |
0 |
0 |
|
In the matrix F the central point appears n0 times. This means that the calculation of coefficient estimates uses the result of each parallel measurement in the centre of the plan and not their average values. Such a construction of the matrix F results from the requirement that all values of the dependent variable included in the matrix Y must have the same variance (Table 18). Furthermore, the parallel experiments in the middle of the schedule allow us to calculate an estimate of the variances of the observation errors.
The significance of the coefficients of the regression equations was assessed using the confidence interval determined in accordance with the values of the Student's test (t-test) and the variance of the parallel experiments. The values of the confidence intervals are the same: for the polynomial of average density Δb1 = 1.2 kg/m3, for the polynomial of strength Δb2 = 9 kPa.
The evaluation of the adequacy of the obtained models (regression equations with only significant coefficients) was done by comparing the calculated values of the Fisher criterion (F criterion) with its table values, which are 19.2. The calculated values of the F-criterion as the ratio of the variance of adequacy to the variance of the parallel experiments are the same: for the polynomial of average density 14.2, for the polynomial of strength 13.7.
Reviewer 2 Report
The manuscript "Modified polyethylene foam for critical environments" by Ter-Zakaryan and co-authors aims to evaluate deferent parameters of building structures for use in extreme conditions.
The manuscript is mainly written as technical report rather that scientific research paper. The manuscript needs to be rewritten with plain English and clear connected paragraphs an sentences.
As there are several parameters under the study it is important to use a multi component analysis like the "Principle Component analysis, PCA".
- "Annotation" to be changed to "Abstract" and it is normally written in a single paragraph format
- in the "Abstract" it is mentioned that "... according to standard methods ..." it is good to mention some of these standard methods used in this work.
- in Material and Discussion and Introduction sections it is mentioned that "TEPOFOL technology" is this a technology or a company name? it is important to have more introduction about this technology for the readers.
- in the Conclusion "Insulation materials based on polyethylene foam are environmentally friendly, easy to use and install, and have high operational stability." are these conclusions based on the data and analysis resulted from this work!
- References and their format is not matching the MDPI formats.
The manuscript in its form is not clear and suitable for Polymers readers.
Author Response
Reviewer 2
- Since several parameters are examined, it is important to use a multicomponent analysis like principal component analysis.
The studies, based on the methods of mathematical design of the experiment, were carried out in two series of experiments. Based on the analysis of scientific sources (articles in professional journals), the factors influencing the response functions and the intervals of their variation in the experiment were determined. A preliminary series of experiments was carried out (using the fractional replicate method), which made it possible to identify the most important factors (the content of secondary polyethylene; the content of modifier, the consumption of flame retardant) and to clarify the intervals of variation of these factors.
2 "Annotation" to be changed to "Abstract" and it is normally written in a single paragraph format
"Abstract" edited.
- in the "Abstract" it is mentioned that "... according to standard methods …" it is good to mention some of these standard methods used in this work.
The studies were conducted in accordance with state standards. On the territory of the Russian Federation and CIS, most of which are updated with international standards. We list the main GOSTs, but doubt that it is appropriate to mention each of them in the article, even in the summary:
GOST EN 1608-2011 "Thermal insulating products for building. Method for determining the tensile strength parallel to the faces.
GOST EN 12088-2011 "Thermal insulation products for building. Method for determination of diffusion moisture absorption over a long period of time";
GOST EN 1609-2011 "Thermal insulating products for construction. Method for the determination of water absorption during short term partial immersion".
GOST 17177-94 "Thermal insulating building materials and products".
GOST EN 1606-2011 "Thermal insulating products for construction. Method for determination of creep under pressure.
GOST R 56729-2015 (EN 14313:2009). "Polyethylene foam products, thermally insulating, factory-made, for the technical equipment of buildings and industrial plants. General technical conditions".
GOST 30244-94. Building materials. Flammability test methods.
- In Material and Discussion and Introduction sections it is mentioned that "TEPOFOL
technology"… is this a technology or a company name? it is important to have more
introduction about this technology for the readers.
This is both the name of the company (TEPOFOL LLC) and the name of the technology developed by this company, which is based on the principle of forming seamless insulation shells based on rolled polyethylene foam with a metallised coating, which are used in insulation systems for various building structures
- In the Conclusion "Insulation materials based on polyethylene foam are environmentally
friendly, easy to use and install, and have high operational stability" are these conclusions
based on the data and analysis resulted from this work!
This information is based both on the analysis of a priori information and on the results presented in this article.
- References and their format is not matching the MDPI formats.
Links checked, inconsistencies fixed.
- The manuscript in its form is not clear and suitable for Polymers readers.
The authors are aware that the readership of the journal "Polymers" primarily includes scientists and specialists in the field of polymer synthesis and its application in various fields: from medicine to aerospace. At the same time, as the experience of the December 2021 special issue showed, readers are also interested in the use of certain types of polymer-based products and in the technologies used in construction. At the same time, the circle of readers of the magazine is expanding.
This article is a logical continuation of material previously published in Polymers magazine: Ter-Zakaryan K. A., Zhukov A. D., Bobrova E. Yu., Bessonov I. V., Mednikova E. A. Foam Polymers in Multifunctional Insulating Coatings // Polymers 2021, 13(21), 3698; https://doi.org/10.3390/polym13213698 [References, 17]
Reviewer 3 Report
The manuscript is very much interesting from waste management point of view. Lot of effort has been put off to demonstrate those results. The article opens new window of for recycled polyethylene foam in cold polar regions.
The article should get published after the revisions mentioned below.
Comment 1: Which statistical software tool was used to do the optimization?
Comment 2: Why differential analytical method was used for the optimization and for designing the experimental design? Statistically significance data is missing like p value or ANNOVA.
Comment 3: Prepared Material property analysis is missing like prepared foam surface morphology and foam network dimension analysis.
Author Response
Reviewer 3
Comment 1: Which statistical software tool was used to do the optimization?
Analytical optimization is a deterministic method (see answer to comment 2).
Comment 2: Why differential analytical method was used for the optimization and for designing the experimental design? Statistically significance data is missing like p value or ANNOVA.
In the studies the method of analytical optimisation was used, absolutely rightly, based on differential analysis). This method was developed at the Moscow University of Civil Engineering and can significantly reduce the processing time for the results of the experiment and the material costs for conducting the experiment. This method has already been used in the analysis of dozens of technologies and formulations of various building materials.
In applying the method of analytical optimisation, the regression equations (with confirmed adequacy in terms of strength and average density) were considered as algebraic functions of several (in this case three) variables to which methods of mathematical analysis are applicable, including the search for the optimum of the function by defining partial derivatives.
The adequacy of the models obtained was checked using the Fisher criterion. Additions were made to the article.
Comment 3: Prepared Material property analysis is missing like prepared foam surface
morphology and foam network dimension analysis.
The treatment (processing) of secondary polyethylene includes the following technological steps: collection and storage, shredding and agglomeration, removal of foreign inclusions, addition of modifiers, including flame retardants, granulation. If waste from own production (offcuts, low-quality products) is used, it is granulated after being processed by pyrolysis and the addition of modifiers and then dosed into the feed hopper of the extruder.
The experiment is based on the use of a cybernetic "black box" model in which all internal interactions between the individual components of the object under study are not taken into account. Only external influences (of variable factors) and the results of these influences (reaction functions) are examined.
A de facto study of the properties of modified polyethylene foam using recycled polyethylene has shown that they differ slightly from the properties of a material based on pure synthetic raw materials, including the size of the polyethylene foam cells (up to 1.2 mm).
Round 2
Reviewer 1 Report
The authors have addressed the problem very well, and the manuscript can be accepted in the present form.
Reviewer 2 Report
The manuscript has been improved and the comment points become clear.
Reviewer 3 Report
Now it's acceptable to get published.